# Shadow Knowledge Distillation: Bridging Offline and Online Knowledge Transfer

**Lujun Li**[1,2,✉]**, Jin Zhe**[1,✉]
School of Artificial Intelligence, Anhui University, China
Chinese Academy of Science, China
lilujunai@gmail.com; jinzhe@ahu.edu.cn

## Abstract

Knowledge distillation can be generally divided into offline and online categories according to whether teacher model is pre-trained and persistent during the distillation process. Offline distillation can employ existing models yet always demonstrates inferior performance than online ones. In this paper, we first empirically show that the essential factor for their performance gap lies in the reversed distillation from student to teacher, rather than the training fashion. Offline distillation can achieve competitive performance gain by fine-tuning pre-trained teacher to adapt student with such reversed distillation. However, this fine-tuning process still costs lots of training budgets. To alleviate this dilemma, we propose SHAKE, a simple yet effective **SHA**dow **K**nowl**E**dge transfer framework to bridge offline and online distillation, which trades the accuracy with efficiency. Specifically, we build an extra shadow head on the backbone to mimic the predictions of pre-trained teacher as its shadow. Then, this shadow head is leveraged as a proxy teacher to perform bidirectional distillation with student on the fly. In this way, SHAKE not only updates this student-aware proxy teacher with the knowledge of pre-trained model, but also greatly optimizes costs of augmented reversed distillation. Extensive experiments on classification and object detection tasks demonstrate that our technique achieves state-of-the-art results with different CNNs and Vision Transformer models. Additionally, our method shows strong compatibility with multi-teacher and augmentation strategies by gaining additional performance improvement. Code is made publicly available at https://lilujunai.github.io/SHAKE/.

## 1 Introduction

Even though Deep Neural Networks (DNNs) have great success in tackling a variety of challenges over computer vision [27, 13] and natural language processing [2], they usually have large numbers of parameters, bringing heavy computation costs. To alleviate this problem, many network compression methods [29, 46, 6, 21, 9] have been proposed, among which Knowledge Distillation (KD) [19, 58, 14] has recently attracted increased attention.

KD aims to transfer knowledge from a high-capacity large model (*i.e.*, teacher) to a low-capacity lightweight model (*i.e.*, student). Numerous offline methods [1, 19] use a two-stage training process that begins with training a teacher model and then keeping it fixed to distill student model (see KD in Figure 1 (a)). Besides offline fashion, recent online methods [28, 5] adopt a one-stage training process, jointly training the student and teacher/peer models using bidirectional distillation like DML [66] in Figure 1 (b). These online distillations always surpass the offline distillations under the same teacher model. However, some large teacher models trained from scratch would bring some difficulties (*e.g.*, high computational resources and unstable optimizations) for distillation, especially for tasks that rely on large transformers (*e.g.*, BERT [57], GPT-3 [2] and ViT-MoE [48]). Therefore, two questions

36th Conference on Neural Information Processing Systems (NeurIPS 2022).

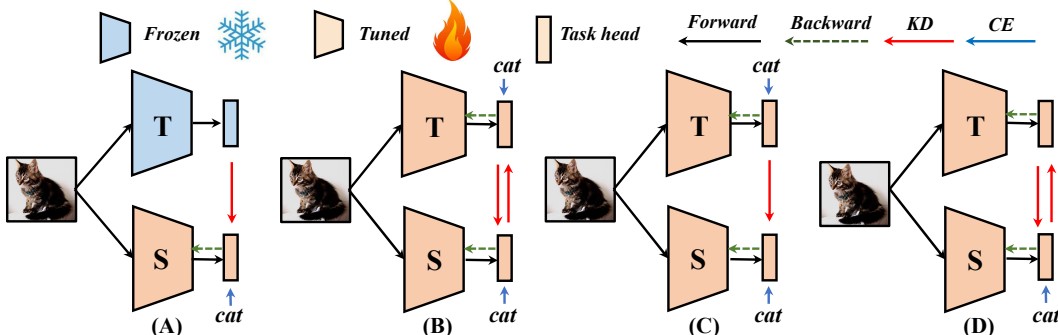

Figure 1: Illustration of (a) KD, (b) DML, (c) DML† without $KD_{S \to T}$, and (d) KD† with $KD_{S \to T}$. $KD_{S \to T}$ denotes reversed distillation from student to teacher.

Table 1: Left: comparison of training time, Top-1 accuracy (%), and teacher-student gap (T-S gap) among the (a) KD, (b) DML, (c) DML† without $KD_{S \to T}$, (d) KD† with $KD_{S \to T}$, and our SHAKE for ResNet-20 (69.09%) via pre-trained teacher (Pre-T) ResNet-110 (74.31%) on CIFAR-100. Training time is measured on a single 2080Ti GPU, and × represents the improving ratios than KD. The teacher-student gap [8] is defined as KL divergence between their outputs (lower is better). Right: training time and accuracy of different settings for KD† and SHAKE. KD† (head) means only updating head, and KD† (adapter) is performed by adding adaptation layers (Conv-BN-ReLU) for output of each stage of teacher and fine-tuning adaptation layers. SHAKE (share) denotes the proxy teacher shares the backbone. SHAKE (multi) denotes SHAKE via multi-ResNet-110 settings [51].

| Method | Pre-T | $KD_{T \to S}$ | $KD_{S \to T}$ | Time | Top-1 | T-S gap | | Method | Top-1 | T-S gap |
|---|---|---|---|---|---|---|---|---|---|---|
| (A) KD | ✓ | ✓ | ✗ | ×1.00 | 70.66 | 1.12 | | (F) KD† (head) | 71.05 | 1.05 |
| (B) DML | ✗ | ✓ | ✓ | ×4.32 | 71.52 | 0.38 | | (G) KD† (adapter) | 71.34 | 0.92 |
| (C) DML† | ✗ | ✓ | ✗ | ×4.41 | 70.55 | 0.82 | | (H) SHAKE (alone) | 71.82 | 0.45 |
| (D) KD† | ✓ | ✓ | ✓ | ×4.29 | 71.76 | 0.66 | | (I) SHAKE (share) | 72.02 | 0.51 |
| (G) SHAKE | ✓ | ✓ | ✓ | ×1.28 | 72.02 | 0.51 | | (J) SHAKE (multi) | 72.35 | 0.58 |

naturally arise: **(1) Why is there a performance gap between offline and online distillation? (2) How the performance of offline distillation can be advanced with design techniques of online KD methods?**

To clarify the first question, we compare KD and DML regarding the training fashion of teacher and distillation loss in Table 1. Contrary to the common belief, we empirically observe that training fashion may not affect the distillation performance since DML† obtains similar performance as KD (70.55% vs. 70.66%). Instead, the reversed distillation from the student model yields significant accuracy gains for KD† than KD (71.76% vs. 70.66%) and DML than DML† (71.52% vs. 70.55%).

Then, we analyze the output discrepancy of the teacher-student for different methods, and the reversed distillation reduces the gap from 1.12 of KD to 0.66 of KD†. Therefore, the main reasons behind the performance gap lie in two aspects: (a) Teacher models in conventional offline pipelines are not optimized for the student model. Thus, they could only provide general knowledge, which may be suboptimal for the particular student. (b) The reversed distillation changes the universal teacher model into a student-aware one and bridges the teacher-student gap, which clearly justifies:

*Teachers should teach students in accordance with their aptitude and should not follow the same pattern.*

For the second question, fine-tuning the pre-trained teacher model with reversed distillation is a straightforward way to bridge the two training fashions. However, in most scenarios, fine-tuning the whole network still needs lots of training budges (more 4× costs than KD in Table 1). Some trade-offs in reducing the fine-tuning parameters or adapter-based methods (see Figure 2 (G)) involve performance loss. Thus, how to augment reversed distillation without much extra overhead is an important issue for the application. Besides fine-tuning, building a proxy teacher model (see Figure 2 (H)) to inherit the knowledge of pre-trained models and receive reversed distillation from students also enjoys the same benefits. As shown in Table 1 (H), this proxy teacher model (SHAKE (alone)) can achieve competitive gains than fine-tuning the whole teacher model (KD†). Recent weight-sharing strategies in AutoML [62, 43, 21, 9] can effectively save training overhead and preserve logit

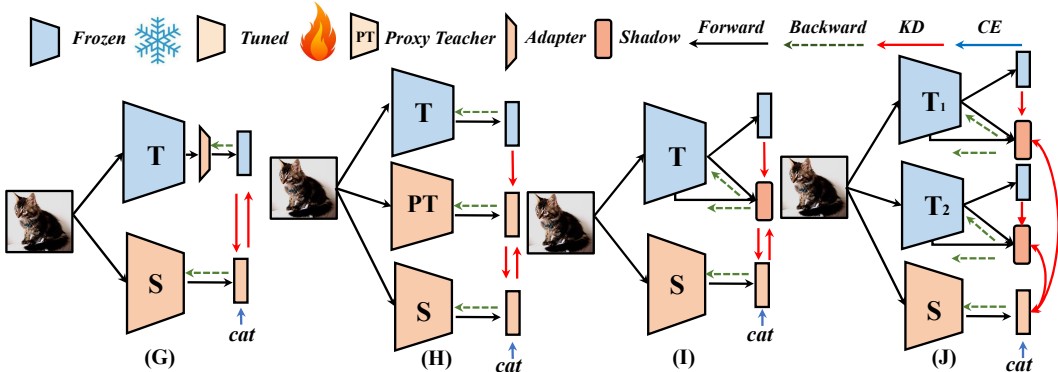

Figure 2: Evolution of our **SHA**dow **K**nowl**E**dge distillation (SHAKE). (G) KD† with an additional adaptation layer for teacher fine-tuning. (H) We build a proxy teacher model to inherit knowledge from pre-trained models as an alternative to costly fine-tuning teacher models. (I) This proxy teacher model could reuse the backbone to reduce training budgets and uses the shadow head to perform bidirectional KD with the student. (J) Our SHAKE for multiple teachers: SHAKE leverages multiple shadow heads to individually follow various teacher models.

consistency. This encourages us to adopt the same student architecture and weight-sharing strategy to generate the proxy teacher model. Thus, we allow this proxy teacher to share the backbone but use an individual shadow head to preserve the diversity of logits representations (see Figure 2 (I)). As shown in Table 1 (I), this sharing strategy in SHAKE (share) presents four benefits: (1) more than $3\times$ training acceleration than DML and fine-tuning the whole teacher model. (2) No need for architecture selection cost for proxy teacher. (3) Additional accuracy gains than individual proxy teacher because knowledge inheritance also directly improves the representation in the weight-sharing process.

Based on the above observations, we propose **SHA**dow **K**nowl**E**dge distillation (SHAKE), a novel and effective logits distillation framework. Our SHAKE builds a proxy teacher model and updates its weights via the original teacher model predictions. In this way, SHAKE enjoys the same benefits with teacher fine-tuning to mine the knowledge of the pre-trained model and can perform bidirectional supervision with the student. To optimize training costs, this proxy teacher shares the backbone with the pre-trained teacher but leverages an individual head to preserve the diversity of logits representations. This head is named shadow head since it imitates the original teacher model just like its shadow. During training, we only need to additionally train this shadow head with small training budgets (only $1.28\times$ costs than KD). After training, all teacher models and shadow module can be deprecated, and student model can be separately applied for inference without any overhead. Moreover, we extend the SHAKE to multi-teacher model scenarios using multiple shadow heads to inherit diverse knowledge.

In principle, SHAKE alters the chain of knowledge transfer from ***pre-trained teacher→student*** in KD to ***pre-trained teacher→proxy teacher⇌student***. Other adaptive KD [40] only employs middle networks to distill as ***large teacher→middle teacher→student***, which is not optimized for students and requires multi-step training. The merits of SHAKE lie in three-fold. First, it effectively reduces the teacher-student capability gap with reversed distillation, bringing significant gains when pre-trained teacher models are available. Second, for the scenario without pre-trained teacher models, SHAKE also enables the offline KD methods to be more effective in alleviating the unstable optimization issues of online KD methods. Thus, SHAKE bridges the offline and online KD methods and enjoys the advantages of both methods. Third, SHAKE achieves favorable trade-offs between accuracy and training budget. By contrast, other adaptive KD [40] needs to sequentially train multiple models with lots of additional training time and resources. We hope that these intriguing observations in SHAKE would expand the application of KD and facilitate future research for KD work to some extent.

We conduct extensive experiments on multiple tasks (*e.g.*,classification and detection) and datasets (*e.g.*, CIFAR-10, CIFAR-100, Tiny-ImageNet, ImageNet, and MS-COCO) to verify the superiority of the proposed method. SHAKE achieves a consistent and significant accuracy boost in various neural networks and data augmentations, which outperforms other methods by large margins. For example, SHAKE obtains $2.92\% \sim 6.75\%$ accuracy gains than baseline and more than $3\times$

training acceleration than DML on CIFAR-100. On the challenging ImageNet dataset, SHAKE can augment the performance of ResNet-18 from 69.75% to 72.07% and MobileNet from 70.13% to 72.66%, which are state-of-the-art results for KD techniques. On vision transformer architecture [55], our approach achieves 75.22% Top-1 accuracy and 2.76% gain for training a VIT-T model. On the object detection task, SHAKE improves the AP by 1.02 for Faster R-CNN on the MS-COCO, demonstrating its generality.

In summary, we make the following principle contributions in this paper:

- By analyzing and exploring the difference between offline and online KD methods, we empirically show that the reversed distillation hinders the performance gain, which fixes the discrepancy between teacher-student capability. This motivates us to propose a new **SHA**dow **K**nowl**E**dge distillation (SHAKE) framework to bridge two training fashions, which, to the best of our knowledge, is not achieved throughout the area of knowledge distillation.
- SHAKE achieves remarkable trade-offs between accuracy and training efficiency with an extra shadow head. The shadow head inherits knowledge from pre-trained models, introduces reversed distillation with students, and accelerates the training process.
- We perform thorough evaluations on classification and detection. SHAKE achieves state-of-the-art performances in various datasets and architectures (*e.g.*, CNN and vision transformer). Specifically, ResNet-18, MobileNet, and VIT-T with SHAKE achieve 72.07%, 72.66% and 75.22% Top-1 accuracy on ImageNet, outperforming KD by 1.41%, 1.98% and 3.02%, respectively.

## 2 Shadow Knowledge Distillation

In this section, we first revisit the offline and online KD methods. Then, we present the formulation of SHAKE and understand why SHAKE work from the perspective of optimization properties and teacher-student similarity. Finally, we present the formulation of SHAKE for multi-teacher models. The evolution of our approach is shown in Figure 2.

### 2.1 Revisit of offline and online KD methods

We first review the formulations of offline and online KD. For simplicity, We choose two typical frameworks (*i.e.*, original KD [19] and DML [66]) for analysis. Regarding training data $(X, Y)$ containing training samples $X$ and labels $Y$. Let $f_T$ be the output logits of the fixed teacher $T$ and let $f_S$ be the output of student $S$, respectively. KD equips the student network $f_S$ via minimizing:

$$\mathcal{L}_S = \mathcal{L}_{CE}(f_S, Y) + \lambda \mathcal{L}_{KL}(f_S, f_T), \tag{1}$$

where $\lambda$ is used to balance these two terms. $\mathcal{L}_{CE}$ is the regular cross-entropy objective:

$$\mathcal{L}_{CE}(f_S, Y) = CE(Y, \sigma(f_S)), \tag{2}$$

where $\sigma$ is the softmax function and $CE(\cdot, \cdot)$ is the cross-entropy loss. $\mathcal{L}_{KL}$ in Eq. 1 is the distillation goal for transmitting knowledge from a teacher to a student:

$$\mathcal{L}_{KL}(f_S, f_T) = \tau^2 KL\left(\sigma\left(\frac{f_T}{\tau}\right), \sigma\left(\frac{f_S}{\tau}\right)\right), \tag{3}$$

where $\tau$ is the temperature to generate soft labels and $KL$ represents Kullback-Leibler (KL) divergence. The teacher's probabilistic outputs can be viewed as the soft labels in this distillation loss rather than the one-hot ground-truth labels, making it a modified cross-entropy loss.

DML provides a two-way knowledge transfer technique in contrast to KD's one-way distillation. The probabilistic outputs from both teacher and student networks can be used to direct one another's training. DML interleaves two aims to jointly train the teacher and student networks in an end-to-end manner:

$$\mathcal{L}_T = \mathcal{L}_{CE}(f_T, Y) + \lambda \mathcal{L}_{KL}(f_T, f_S)$$
$$\mathcal{L}_S = \mathcal{L}_{CE}(f_S, Y) + \lambda \mathcal{L}_{KL}(f_S, f_T), \tag{4}$$

where the default value of $\lambda$ is 1 in DML, DML consistently outperforms KD under the same teacher-student pair. The common belief argues that the accuracy gap between DML and KD comes from the additional boost of teacher models under mutual training fashion in DML than fixed teachers in KD. However, the significant gains of KD† than KD challenge this belief because the teacher's accuracy of KD† does not obviously improve, but its teacher-student gap reduces, as shown in Table 1. In summary, introducing reversed distillation is a promising way to realize student-adapted KD methods. We illustrate how SHAKE implements it in a cost-friendly way in the following section.

## 2.2 Formulation of SHAKE

As shown in Figure 2 (I), SHAKE proposes a proxy teacher model with output $f_{T'}$ via shadow head to augment reverse distillation in the offline framework. In particular, the hierarchical features $(H_{T_1}, H_{T_2}, ..., H_{T_i})$ of the pre-trained teacher model are transformed by feature adaptation layer $\mathcal{A}$ and then performed via a bottom-up feature fusion. For example, for the three-layer feature fusion, the proxy teacher's output $f_{T'}$ can be defined as:

$$f_{T'} = \pi \times \left\| MLP_{shadow}\left[ \left(\mathcal{A}_1(H_{T_1}) \oplus \mathcal{A}_2(H_{T_2})\right) \oplus \mathcal{A}_3(H_{T_3})\right]_{global-pooling} \right\|_{norm}, \quad (5)$$

where $MLP_{shadow}$ is the fully connected layer with the same shape as the pre-trained teacher. The shadow feature fusion scheme consists of directly feature addition $\oplus$ and Conv-BN-ReLU as feature adaptation layer $\mathcal{A}$. In the backpropagation, pre-trained teachers are fixed, and only shadow modules are updated. Such shadow modules are simple to implement, and its additional costs are similar to other feature distillations [25, 7] and parameter efficient fine-tuning strategies [23]. To achieve better logit alignment, we perform normalization $\| \cdot \|_{norm}$ and scaling $\pi$ for the output of the pre-trained teacher, proxy teacher, and student model, which follows advanced distillation [52, 28, 31].

During the training process, $f_{T'}$ is updated with the output of the pre-trained teacher model $f_T$ and performs mutual distillation with the student models $f_S$ as:

$$\begin{aligned} \mathcal{L}_{T'} &= \mathcal{L}_{KL}(f_{T'}, f_T) + \lambda\mathcal{L}_{KL}(f_{T'}, f_S) \\ \mathcal{L}_S &= \mathcal{L}_{CE}(f_S, Y) + \lambda\mathcal{L}_{KL}(f_S, f_{T'}), \end{aligned} \quad (6)$$

Each loss item has the same form as KD in Eq. 1, and the effects of $\lambda$ are explored in the ablation study. The inherit distillation loss term $\mathcal{D}(f_{T'}, f_T)$ is presented to imitate the probabilistic outputs of pre-trained teacher, which can be defined as KL or MSE distance. The reverse distillation term from student feedback to the proxy teacher and the direct distillation from the proxy teacher to the student play key roles in our framework, which is analyzed in detail from the perspective of training dynamics and teacher-student similarity in the following.

**Comparison of SHAKE with other KDs in training dynamics**. Eq. 6, Eq. 4 and Eq. 1 clearly illustrate the difference of SHAKE and KD/DML. Compared to KD, SHAKE introduces reversed distillation so that the teacher can be optimized by the student. In contrast to DML, SHAKE leverages the knowledge of pre-trained models, resulting in additional accuracy gains. As shown in Figure 3, SHAKE achieves a robust boost than KD and baseline during the training process. Meanwhile, the training curve of DML shows a highly dynamic oscillation due to unreliable predictions of its teacher trained from scratch. Other adaptive KD methods use the same knowledge transfer way as KD and do not optimize teacher with feedback supervision of student. Thus, these methods, including ATKD [40] and ESKD [8] do not essentially address the teacher-student capacity disparity, and our SHAKE surpasses them with significant margins ($1.18\% \sim 1.29\%$ on ImageNet in Figure 3).

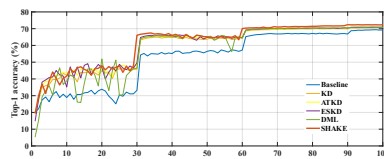

Figure 3: Comparison of training curves of baseline (69.75%), KD (70.66%), ATKD (70.78%), ESKD (70.89%), DML (71.13%) and our SHAKE (72.07%) for ResNet-18 with single ResNet-34 as teacher on ImageNet.

**In-depth analysis about teacher-student similarity in SHAKE.** The similarity between student and teacher network is an important measure for KD tasks. Reversed distillation optimizes the teacher model to adapt to the student, reducing the teacher-student gap and improving distillation efficiency. With reversed KD loss, SHAKE surpasses KD and DML with a small teacher-student discrepancy. To verify this insight, we employ KL-divergence as metrics of similarity [8], where lower KL-divergence implies higher similarity. Figure 4 presents the similarities and performances between the outputs of

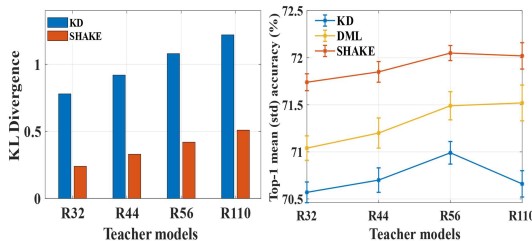
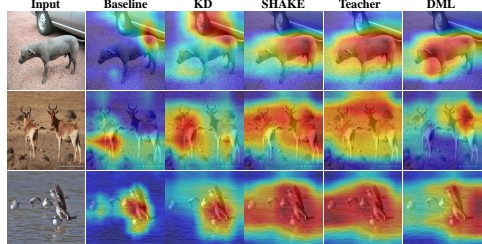

Figure 4: KL-divergence and Top-1 accuracy (%).

Figure 5: Grad-CAM++[4] visualization.

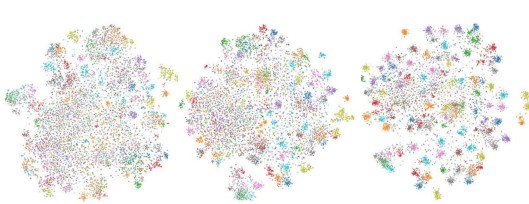
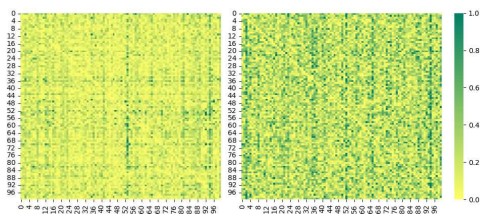

Figure 6: The penultimate layer visualization of ResNet-20 (student) with KD (left), SHAKE (middle) and the teacher (right) on CIFAR-100.

Figure 7: Logits correlation visualization of teacher-student for the student (ResNet-20) with KD (right) and SHAKE (left) on CIFAR-100.

ResNet with different depths as teachers (*e.g.*, ResNet-32, ResNet-110) and student (*e.g.*, ResNet-20). The results show that the distillation from SHAKE always gives a higher similarity than KD and DML, resulting in significant gains under teacher models of different depths. Figure 5 presents the Grad-CAM visualization of ResNet-18 with different methods on ImageNet. The Grad-CAM maps of SHAKE pay more attention to the important regions than KD and baseline, which have high similarity with the teacher. In summary, SHAKE bridges KD and DML to obtain student-friendly distillations, improving teacher-student similarity. In addition, Figure 6 illustrates that comparing with KD, applying SHAKE training helps learn more scattered embeddings [41, 53, 39, 3] and logits correlation visualization in Figure 7 clearly demonstrate that teacher model in SHAKE owns student-aware logits.

## 2.3 Extension to multiple teacher models

As shown in Figure 2 (J), we build several proxy teachers with outputs $(f_{T'_1}, f_{T'_2}, ..., f_{Ti'})$ to follow outputs $(f_{T_1}, f_{T_2}, ..., f_{T_i})$ of multiple teacher models with various shadow heads. Mutual distillation also exists between the task head of the student and these multiple shadow heads. Similar to Eq. 6, the total optimization function for the multi-teacher scenario can be defined as:

$$\mathcal{L}_{T'} = \sum_{i=1}^{N} \mathcal{L}_{KL}(f_{T'_i}, f_{Ti}) + \lambda \sum_{i=1}^{N} \mathcal{L}_{KL}(f_{T'_i}, f_S)$$

$$\mathcal{L}_S = \mathcal{L}_{CE}(f_S, Y) + \lambda \sum_{i=1}^{N} \mathcal{L}_{KL}(f_S, f_{T'_i}). \tag{7}$$

where $N$ is the sum of the total teachers. These multiple teacher models and shadows are present only in the training phase and do not bring extra overhead for inference. As shown in Setting (J) of Table 1, this multi-shadow head strategy yields significant accuracy gains with small additional overhead. Furthermore, SHAKE can be combined with advanced multi-teacher methods (*e.g.*, adaptive weights [51] for different teachers in Table 2).

.

## 3 Experiments

In this part, we first assess our SHAKE for classification and object detection. After that, extensive ablations are conducted to investigate the key our design in SHAKE. As a novel offline logit approach,

the main competitor of SHAKE is the original KD [19]. Thus, we perform detailed experimental comparisons between them. Moreover, we compare the performance with recent advanced KD methods. We employ the open-source versions of these methods with the same training settings for fair comparisons. We choose $\lambda$ and $\tau$ to be 1 and 4 in SHAKE, respectively.

Table 2: Comparison of performances with powerful distillation techniques using the same 240 training epochs. Most results of other methods reference the CRD [54]. SHAKE† denotes SHAKE via multi-teacher settings [51]. W40-2, R32x4, R8x4, R50, MV2, SV1 and SV2 stand for WRN-40-2, ResNet32x4, ResNet8x4, ResNet50, MobileNetV2, ShuffleNetV1 and ShuffleNetV2.

| | Same architectural style | | | | | Different architectural style | | |
|---|---|---|---|---|---|---|---|---|
| Teacher | W40-2 | R56 | R110 | R32×4 | VGG13 | VGG13 | R50 | W40-2 |
| Student | W16-2 | R20 | R20 | R8×4 | VGG8 | MV2 | VGG8 | SV1 |
| Teacher | 75.61 | 72.34 | 74.31 | 79.42 | 74.64 | 74.64 | 79.34 | 75.61 |
| Student | 73.26 | 69.06 | 69.06 | 72.50 | 70.36 | 64.60 | 70.36 | 70.50 |
| FitNets [49] | 73.58 | 69.21 | 68.99 | 73.50 | 71.02 | 64.14 | 70.69 | 73.73 |
| SP [56] | 73.83 | 69.67 | 70.04 | 72.94 | 72.68 | 66.30 | 73.34 | 74.52 |
| RKD [42] | 73.35 | 69.61 | 69.25 | 71.90 | 71.48 | 64.52 | 71.50 | 72.21 |
| CRD [54] | 75.48 | 71.16 | 71.46 | 75.51 | 73.94 | 69.73 | 74.30 | 76.05 |
| ReviewKD [7] | 76.12 | 71.89 | N/A | 75.63 | N/A | N/A | N/A | 77.14 |
| NORM [61] | 75.65 | 71.35 | 71.55 | 76.49 | 73.95 | 68.94 | N/A | 77.06 |
| SRRL [22] | 75.49 | 70.86 | 70.78 | 75.71 | 73.31 | N/A | N/A | 74.18 |
| T$f$-FD [30] | 74.33 | N/A | 70.62 | 73.62 | 71.62 | N/A | N/A | N/A |
| ONE [28] | 74.68 | 70.43 | 70.77 | N/A | 72.01 | 66.26 | 74.35 | N/A |
| KDCL [16] | 74.25 | 70.58 | 70.36 | 74.03 | 71.26 | 65.76 | 73.03 | 74.79 |
| MetaDistil [68] | N/A | 71.25 | 71.40 | N/A | 73.65 | N/A | 74.42 | 77.06 |
| DML [66] | 75.33 | 71.48 | 71.52 | 74.30 | 73.64 | 68.52 | 74.22 | 75.58 |
| DML† | 74.83 | 70.24 | 70.55 | 73.15 | 72.86 | 66.30 | 73.34 | 74.52 |
| KD [19] | 74.92 | 70.66 | 70.66 | 73.33 | 72.98 | 67.37 | 73.81 | 74.83 |
| **KD†(ours)** | **75.78** | **71.62** | **71.76** | **74.91** | **73.85** | **68.81** | **74.10** | **76.42** |
| **SHAKE(ours)** | **76.62** | **72.04** | **72.02** | **77.35** | **74.84** | **70.03** | **74.76** | **77.25** |
| Std & Gain | 0.34/3.36 | 0.12/2.98 | 0.08/2.96 | 0.28/4.85 | 0.38/4.48 | 0.28/5.43 | 0.28/4.40 | 0.28/6.75 |
| KD+FitNets | 75.12 | 71.12 | 71.24 | 74.66 | 73.49 | 67.73 | 73.91 | 77.42 |
| **SHAKE+FitNets** | **76.91** | **72.00** | **72.15** | **78.06** | **74.78** | **70.38** | **75.27** | **78.04** |
| KD+CRD | 75.89 | 70.90 | 71.60 | 75.46 | 74.08 | 69.94 | 74.22 | 76.27 |
| **SHAKE+CRD** | **77.17** | **72.29** | **71.87** | **76.57** | **74.65** | **70.04** | **75.22** | **77.61** |
| KD+Mixup | 75.28 | 71.66 | 71.33 | 75.20 | 74.07 | 67.31 | 73.91 | 76.49 |
| **SHAKE+Mixup** | **76.91** | **71.82** | **72.07** | **77.39** | **75.53** | **70.25** | **75.66** | **78.17** |
| KD+CutMix | 75.66 | 70.90 | 70.69 | 75.39 | 74.78 | 66.39 | 75.04 | 77.44 |
| **SHAKE+CutMix** | **76.29** | **70.92** | **70.90** | **78.28** | **75.11** | **69.44** | **75.98** | **78.27** |
| KD+AVER | 75.22 | 71.08 | 71.24 | 74.99 | 74.90 | 68.91 | 73.26 | 76.30 |
| **SHAKE+AVER** | **76.82** | **72.28** | **72.22** | **78.59** | **75.60** | **70.35** | **75.51** | **77.52** |
| KD+AEKD | 75.68 | 71.25 | 71.36 | 74.75 | 74.75 | 68.39 | 73.11 | 76.34 |
| **SHAKE+AEKD** | **76.88** | **72.32** | **72.35** | **78.90** | **76.26** | **70.42** | **75.67** | **77.60** |

## 3.1 Experiments on CIFAR-100

**Implementation**. With CRD's settings [54], whose training epochs are 240, we perform experiments on several teacher-student models on CIFAR-100, either using the same architecture style or a different one. We employ a conventional SGD optimizer with a weight decay of 0.0005 and a mini-batch size of 64. Initialized at 0.05, the multi-step learning rate decrements by 0.1 at 150, 180, and 210 epochs.

**Comparison with offline and online KD methods**. In Table 2, we compare our approach to some advanced offline/online KD methods with the same training settings. For the same architecture style of the teacher-students, SHAKE obtains $3.35\% \sim 5.45\%$ absolute accuracy gains and outperforms KD with $0.26\% \sim 2.44\%$ margins and Review with $1.72\%$ margins. Besides, SHAKE achieves more significant gains on cross-architecture teacher-student pairs with $4.70\% \sim 6.75\%$ margins than

Table 3: Top-1 and Top-5 mean±std accuracies (%) on ImageNet dataset. Results of Teacher (T), Student (S), KD [19], ESKD [8], ATKD [40], ONE [28], DML [66], CRD [54], AT [64], RKD [42], and OFD [18] are reproduced or refers to CRD [54].

| T | S | Acc | T | S | KD | ESKD | ATKD | ONE | DML | CRD | SHAKE | SHAKE† |
|---|---|-----|---|---|-----|------|------|-----|-----|-----|-------|--------|
| R34 | R18 | Top-1 | 73.40 | 69.75 | 70.66 | 70.89 | 70.78 | 70.55 | 71.03 | 71.17 | **72.07**±0.31 | **72.53**±0.15 |
|  |  | Top-5 | 91.42 | 89.07 | 89.88 | 90.06 | 89.99 | 89.59 | 90.28 | 90.32 | **91.05**±0.22 | **91.26**±0.25 |

| T | S | Acc | T | S | KD | AT | RKD | OFD | DML | CRD | SHAKE | SHAKE† |
|---|---|-----|---|---|-----|-----|------|------|-----|-----|-------|--------|
| R50 | MV1 | Top-1 | 76.16 | 70.13 | 70.68 | 70.72 | 71.32 | 71.25 | 71.13 | 71.40 | **72.66**±0.35 | **73.02**±0.32 |
|  |  | Top-5 | 92.86 | 89.49 | 90.30 | 90.03 | 90.62 | 90.34 | 90.22 | 90.42 | **91.35**±0.25 | **91.62**±0.21 |

Table 4: Top-1 mean±std accuracies (%) for VIT-T with SHAKE on ImageNet dataset.

| Teacher | Student | Tf-KD [63] | Soft KD [55] | Hard KD [55] | DeiT [55] | SHAKE | SHAKE† |
|---------|---------|-----------|--------------|--------------|-----------|-------|--------|
| RegNetY-16GF (82.90) | ViT-T (72.20) | 72.35 | 72.20 | 74.30 | 74.50 | **75.22**±0.10 | **75.72**±0.25 |

baseline. Compared to other KD methods, SHAKE outperforms KD with $0.66\% \sim 1.22\%$ margins, which illustrates the effectiveness of SHAKE in reducing the teacher-student network gap. Compared to DML and DML† under the same teacher-student pair in Table 2, SHAKE obtains $1.79\% \sim 4.20\%$ relative gains. Compare to online KDs (*e.g.*, KDCL [16], ONE [28]) with multiple branches as teacher models, SHAKE also obtains $0.71\% \sim 3.77\%$ relative accuracy gains. These significant improvements demonstrate the superiority of SHAKE by leveraging the knowledge of pre-trained models.

**Orthogonal to other KDs and data augmentations**. As acting on the output logits, SHAKE is orthogonal to feature and relation KD methods because of transferring different knowledge. As shown in Table 2, the combination of FitNets with SHAKE surpasses KD with $0.91\% \sim 2.65\%$ margins. In addition, for CRD, its combination with SHAKE yields more dramatic gains than KD. For distillation training under strong data augmentation (*e.g.*, Mixup and CutMix [50]), SHAKE obtains $0.21\% \sim 3.05\%$ relative accuracy gains than KD.

**Extension to multiple teacher models.** In Table 2, we compare SHAKE and KD with average and adaptive weighting in multiple teacher settings. The results (*i.e.*, $0.75\% \sim 4.05\%$ gains than KD) illustrate that the design of SHAKE with the multi-proxy model can effectively inherit the knowledge of different teachers bringing significant performance gains.

## 3.2 Experiments on ImageNet

**Implementation**. For the standard ResNet-18 [17] and MobileNet [20] models, we employ the same training configurations, whose the training epochs are 100, as most distillation techniques. The learning rate is initialized at 0.1, decreasing by 0.1 every 30 epochs. Recent vision transformers achieve great success on different vision tasks [10, 55]. We also extend SHAKE to VIT-T [55] with the same training settings (*e.g.*, data augmentation and distillation token) on ImageNet. Implementation details are available in supplementary materials.

**Comparison results**. Table 3 reports the performance of our approach on ImageNet. SHAKE improves the baseline models of ResNet-18 with 2.32% gains in Top-1 accuracy (see Figure 3 for detailed accuracy curves) and MobileNet with 2.53% gains. Compared to other KD methods, SHAKE outperforms KD with $1.41\% \sim 1.98\%$ margins and CRD with $0.90\% \sim 1.26\%$ margins, which supports the superiority of SHAKE on the large-scale dataset. Equipped with the distillation of two teacher models, SHAKE† obtains 2.78% gain for ResNet-18 and 2.89% gain for MobileNet than baseline. As shown in Table 4, SHAKE obtains 2.76% accuracy gains than baseline. It surpasses KD under the soft or hard label of the CNN teacher (RegNet [45]), verifying its effectiveness on different architectures.

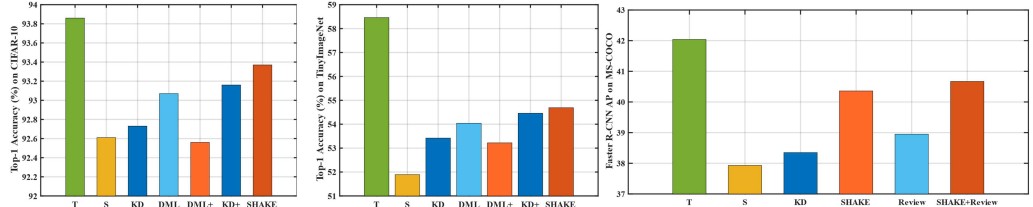

Figure 8: Left: results for ResNet-20 as Student [S] with ResNet-110 as Teacher [T] on CIFAR-10. Middle: results for ResNet-20 [Student] with ResNet-110 [Teacher] on Tiny-ImageNet. Right: results of Faster R-CNN-R50 [Student] via Faster R-CNN-R101 [Teacher] on MS-COCO.

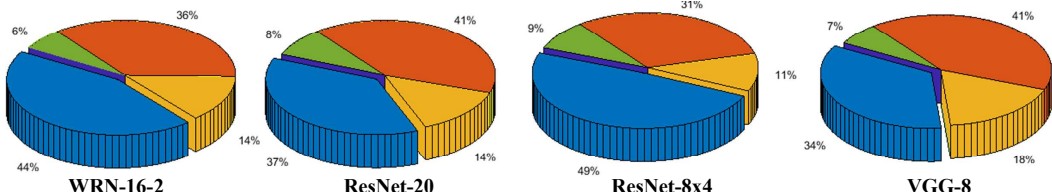

Figure 9: Relative gain ratios in SHAKE of reversed distillation (blue part), shadow designs (yellow part), inherited distillation from pre-trained teacher (green part) and distillation to student (red part) for different models in Table 2 on CIFAR-100.

## 3.3 Extensive experiments in diverse datasets and tasks

**Experiments on CIFAR-10 and Tiny-ImageNet.** In addition to the commonly evaluated classification datasets (*i.e.*, CIFAR-100 and ImageNet) in knowledge distillation methods, we also evaluated SHAKE in CIFAR-10 and Tiny-ImageNet, which have smaller categories and data volumes. As shown in Figure 8, our intriguing observation about Reversed distillation is also present on different data sets: KD† with reversed KD outperforms KD, DML, and KD†. In addition, with a carefully designed shadow module and distillation strategy, our SHAKE outperforms KD† and achieves best results.

**Experiments on object detection**. Besides classification tasks, we also extend SHAKE for object detection applications. In particular, we evaluate SHAKE on MS-COCO dataset [35] and use the most popular open-source Detectron2 [34] as the strong baseline. We apply SHAKE to two-stage detector (*i.e.*, Faster R-CNN [47]), which are widely used object detection frameworks. Following common practice [34, 7], all models are trained with a $2\times$ learning schedule (24 epochs). As shown in Figure 8, our SHAKE improves the AP 1.02 on Faster R-CNN, which outperforms KD [19]. The performance of object detection greatly depends on the quality of deep features to locate interesting objects, while logits are not capable of providing knowledge for object localization. Thus, SHAKE is naturally weaker than Review [7] (recent excellent feature-based KDs), and we further introduce Review to obtain satisfactory results. It can be observed that SHAKE can obtain new best results with feature-based KDs. The success of challenging object detection tasks indicates the generality of our approach and orthogonality for the det-distillers [59, 65].

## 3.4 Ablation study

**Detailed ablation study of SHAKE** As shown in Figure 9, an ablation study has been conducted to demonstrate the individual effectiveness of different components in SHAKE. It is observed that (a) Reversed distillation optimizes the teacher model to adapt to the student, improving distillation efficiency. SHAKE, without reversed distillation, only obtains marginal gains. (b) Shadow modules effectively preserve the diversity of knowledge, and their backbone-sharing settings can accelerate the student model's convergence and result in performance gains. In addition, the stronger shadow module design for storing and teaching more pre-trained knowledge when the capacity gap enlarges. (c) Inherited knowledge from pre-trained teacher via distillation optimization beyond sharing weights is important. Proxy teachers in our SHAKE transfer the knowledge of the original pre-trained teacher model more efficiently. These results demonstrate that our proxy teacher design can indeed serve as a

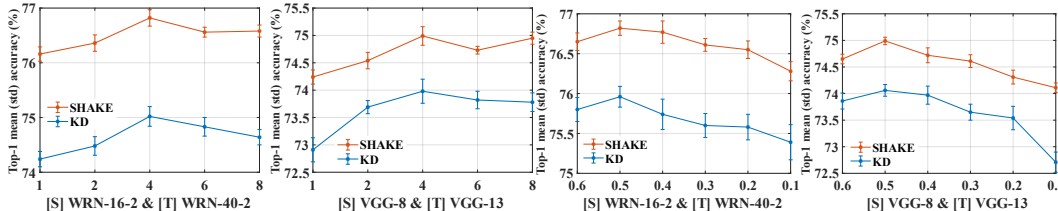

Figure 10: Top-1 mean (std) accuracies (%) of varying $\tau$ for WRN-16-2 & VGG-8 (left) and varying $\lambda$ for WRN-16-2 & VGG-8 (right) on CIFAR-100.

good bridge between the original teacher and student. Reverse KD reduces the teacher-student gap to facilitate knowledge transfer.

**Sensitivity study for temperature $\tau$ and weight $\lambda$.** $\tau$ in Eq. 3 controls the softness of teacher's supervision. As $\tau$ increases, the output of the softmax function becomes smoother. As shown in Figure 10, SHAKE presents superior gains and robustness than the original KD in different $\tau$. $\lambda$ balances the implications of KD loss. The small $\lambda$ limits distillation gains, and $\lambda = 0.5$ is the best option for SHAKE.

## 4   Related Work

**Offline distillations**. In the offline framework [19], the teacher is pre-trained and non-updated, and then its soft logits are used as extra supervision to distill students. Although subsequent methods explored to transfer feature knowledges [49, 64] and relation knowledges [42, 54]), the effective original KD still outperforms most distillations and is widely used for different tasks [12, 26, 31, 32]. However, the pre-trained teacher model is not optimized for the student, limiting its gains. To address this issue, our SHAKE enhances KD using a shadow head to introduce optimization from the student.

**Online distillations**. Online KD methods simplify the KD process by training all models simultaneously. DML [66] performs bidirectional distillation for the peer networks and ONE [28] presents an on-the-fly ensemble distillation among multiple branches. Subsequent studies focus on how to balance multiple teacher [5] or construct the online teacher [60, 33, 38, 33]. Online KD methods always obtain better performance than offline ones. However, large teacher models trained from scratch sometimes perform unstable predictions and bring lots of training costs during distillation. Therefore, we propose that SHAKE combine the advantages of the two pipelines to facilitate application.

**Adaptive distillations**. Capacity gaps between teacher-student models for their different architectures would limit distillation gains [37]. There are two types of existing works to alleviate this gap in terms of training paradigms [11] and architectural adaptation [24, 15]. For instance, ESKD [8] proposes stopping the training of the teacher early, and ATKD [40] uses a medium-size teacher assistant to perform sequence distillation with large overheads. However, these methods do not optimize the teacher model for the student, resulting in minor benefits. Recent Meta-KD [68, 67, 36] implement quite complex meta-optimization for whole teacher networks with a lot of extra overhead. In sharp contrast to these methods, SHAKE is a student-aware offline KD method and opens a new direction for adaptive distillation design.

## 5   Conclusion

In this paper, we present SHAKE, a new student-aware logit distillation that is easy to use and effective for bridging offline and online knowledge transfer. Based on our insight into online KD methods success, SHAKE achieves this goal by building an extra shadow head as a proxy teacher model to perform mutual distillation with the student model. Thorough evaluations are performed on classification and detection, and SHAKE achieves significant performance gains in various neural networks without extra inference overheads. In future work, we would make the best effort to explore the application of SHAKE for different specific tasks [44]. We hope that this elegant and practical approach would inspire future research on knowledge distillation design and understanding.

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
