# Supplementary Materials for *Shadow Knowledge Distillation: Bridging Offline and Online Knowledge Transfer*

**Lujun Li**[1,2,✉]**, Jin Zhe**[1,✉]
School of Artificial Intelligence, Anhui University, China
Chinese Academy of Science, China
lilujunai@gmail.com; jinzhe@ahu.edu.cn

## 1 Main Experimental Settings

In this section, we provide detailed settings of the classification experiments and extended experiments.

### 1.1 Experiments on CIFAR

**Dataset**. CIFAR [3] is the most widely used classification dataset for evaluating the performance of distillation methods. It includes 50,000 training and 10,000 test images.

**Implementation**. In the comparison experiments with other offline KD methods, we use the same training settings of CRD [11] to implement various KD methods [5, 6, 4, 7, 12], whose training epochs are 240. We used a $32\times32$ random crop after padding with 4 pixels and a random horizontal flip, and we optimized the models with the stochastic gradient descent (SGD) algorithm with a learning rate of 0.05 and applied learning rate decay in 150, 180, and 210 epochs, for a total of 240 epochs. As a student model, the initial learning rate in ShuffleNetV1 and ShuffleNetV2 is set to 0.01. We used a 5e-4 weight decay, a momentum of 0.9, and a batch size of 64.

### 1.2 Experiments on ImageNet

**Dataset**. We also perform experiments on the ImageNet dataset (ILSVRC12) [10], which is regarded as the most difficult classification task. It has approximately 1.2 million training images and 50,000 validation images, with each image belonging to one of 1,000 categories.

**Implementation**. In the ImageNet experiments, the student models (*i.e.*, ResNet-18 [1] and Mo-bileNet [2]) are trained with 100 epochs. For the data augmentation, we employ the standard data augmentation technique, which includes random cropping, random horizontal flipping, and brightness adjustment. We used the SGD algorithm for the optimizer, with a Nesterov momentum of 0.9, weight decay of 0.0001, and an initial learning rate of 0.1. Other KD methods are implemented using the hyperparameter settings of original paper. The SHAKE's detailed settings are same to the CIFAR-100.

### 1.3 Experiments on vision transformer

**Implementation**. We utilize the same data augmentation and regularization methods described in DeiT for fair comparison (*e.g.*, Auto-Augment, Rand-Augment, mixup). We use AdamW as the optimizer, with a learning rate of 0.001 and a weight decay of 0.05. The entire training procedure consists of 300 epochs. The first five epochs are for warm-up, and the learning rate follows a cosine decay function in the remaining epochs. SHAKE, like DeiT, uses the distillation token with the shadow head as the proxy model. Furthermore, SHAKE incorporates mutual distillation between the shadow head and the classification head, yielding significantly higher gains than DeiT.

36th Conference on Neural Information Processing Systems (NeurIPS 2022).

## 1.4 Experiments on object detection.

**Dataset**. We evaluate SHAKE on MS-COCO dataset [8] , which contains more than 120K images, covering 80 categories. All performance is evaluated on the MS-COCO validation set.

**Implementation**. We apply SHAKE Faster R-CNN [9]) and initialize the backbone with weights pre-trained on ImageNet [10]. Horizontal image flipping is utilized in data augmentation. For SHAKE, we build an extra shadow head with the same architecture as the original classification head, which performs distillation in the fine-tuning detector stage. Following most of the output logits distillation on the detection, our SHAKE conducts distillation on the classification and regression output predictions. For classification, we exploit the KL divergence for multi-label distillation. For regression, our SHAKE minimizes the bounding box distance between teacher-student. The two distillation loss terms are combined with the summation.