# OpenReview forum: "Shadow Knowledge Distillation: Bridging Offline and Online Knowledge Transfer"
_NeurIPS.cc/2022/Conference — NeurIPS 2022 Accept_

### Official Review · Reviewer_txDD · 2022-06-15

**Rating:** 6
**Confidence:** 5
**Soundness:** 2 fair
**Presentation:** 3 good
**Contribution:** 3 good

**Summary:**

This paper shows that the merits of online knowledge distillation (DML) over the offline knowledge distillation come from the fact that the teacher model are optimized for the student model. Then, to reduce the training overhead of fine-tuning the teacher in online knowledge distillation, they propose SHAKE which distills knowledge from the teacher to the proxy teacher offline, and then perform DML between the proxy teacher and the student. Experiments on both detection and image classification are provided to demonstrate their effectiveness.

**Questions:**

Please refer to the weakness.

**Strengths And Weaknesses:**

Strengths.
1. I like the study in Line 37-49, which gives a surprising explanation of the effectiveness of online knowledge distillation.
2. The authors have given very sufficient studies on both classification and object detection, with the combination of the other KD methods, and different data augmentation settings.
3. Good paper writing.

Weakness.
1. About the motivation. In Line44-49, the authors claim that the teacher model in offline knowledge distillation does not work well because it has not been optimized for the student model. I can understand that this conclusion is drawn from the observations in Line41-43. However, even in if DML and the other offline knowledge distillation methods, the "teacher model" is trained to minimize the cross-entropy loss between teacher prediction and the labels, and the KL (of feat loss) between teacher prediction and student predictions. In these situations, the teacher model is also not optimized for the student model. To make it easy to be understood, when the teacher in DML minimizes the knowledge distillation loss, it does not optimize the loss for the student, it just optimizes it for itself. Hence, the problem that why an offline teacher works worse is still not clear.

2. In the proposed SHAKE, the teacher knowledge is firstly transferred to the proxy teacher, and then transferred to the student. Thus, it is similar to the teacher-assistant knowledge distillation (TAKD, AAAI2020)[1].  It will be better to discuss the difference and connection between SHAKE and TAKD.

3. One of the motivations of SHAKE is to reduce the fine-tuning overhead of the pretrained teacher. Recently, adapter-based methods have been proposed to efficiently fine-tune a pretrained model. It will be better to compare SHAKE with using these adapter methods for teacher fine-tuning.

4. In the detection experiments in Table 4., SHAKE+Review leads to 0.31 AP improvements than Review, which is not significant.  Besides, both KD (logit) and Review are not knowledge distillation methods for object detection. It will be better if results with detection-oriented knowledge distillation methods, such as [2-3].


[1] Improved knowledge distillation via teacher assistant. AAAI2020

[2] Improve object detection with feature-based knowledge distillation: Towards accurate and efficient detectors.ICLR2020.

[3] General instance distillation for object detection. CVPR2021.


I am happy to increase my rating if my questions are answered in the rebuttal.

---

> ### Author Response · Authors · 2022-08-02
> **Responses to the review of Reviewer txDD-Part-III**
>
> ---
>
> **Q4:** About the results with detection-oriented knowledge distillation methods.
>
> **A4:** Thanks for the suggestion. Following your suggestion, we evaluate the SHAKE with detection-oriented knowledge distillation methods. The results show that the distillation improvement of SHAKE with Fb-KD (ICLR2020) or GID (CVPR2021) is significantly better than SHAKE + Review.
>
> =======================================================================
>
> Comparison of results of object detection on MS-COCO.
>
> | Model                          | AP    | APL   | APM   | APS   |
> | ------------------------------ | ----- | ----- | ----- | ----- |
> | Faster-R-CNN-R101 [T]          | 42.04 | 54.6  | 45.55 | 25.22 |
> | Faster-R-CNN-R50 [S]           | 37.93 | 49.1  | 41.14 | 22.44 |
> | KD (logits)                    | 38.35 | 49.48 | 41.8  | 22.73 |
> | Review (feature)               | 40.36 | 52.87 | 43.81 | 23.60  |
> | SHAKE (logits)                 | 38.95 | 50.78 | 42.32 | 22.88 |
> | SHAKE & Review                 | 40.67 | 52.96 | 43.92 | 23.85 |
> | SHAKE & Fb-KD (ICLR2020)  [R2] | 41.23 | 53.85 | 44.76 | 23.88 |
> | SHAKE & GID (CVPR2021)  [R3]   | 41.51 | 54.26 | 44.98 | 23.95 |
>
> ---
>
> **Finally,** we hope our response could address the concerns, and we thanks again for the valuable suggestions. We are glad to discuss further comments and suggestions.

---

> > ### Comment · Reviewer_txDD · 2022-08-09
> > **Response to Authors: Keep my rate**
> >
> > The authors have addressed most of my concerns in the rebuttal.
> >
> > About my question-1. Actually, my question is "why a teacher optimized with the student is a better teacher". Reducing the gap between students and teachers may be a reason. But it is not very convincing because the student-teacher gap theory is proposed based on the difference between the number of parameters (in TAKD) while optimizing with the student has no influence on parameters. Besides, authors in TAKD provide theoretical analysis from the perspective of PAC and learning with privileged information. I advise authors to investigate more from this direction, which can improve this paper. The current answer from the author is ok for me but still not perfect. (This advice is not for rebuttal now, just for your future improvements on this paper)
> >
> > About my question-3. Thanks for your reply. It will be better if the training cost can be reported because the advantage of adapter-based methods compared with finetuning is the low training overhead.
> >
> > About my question-4. The performance improvements with Fb-KD and GID are not that significant. But I appreciate your efforts in providing these experiments and hope you can include these results in your final version.
> >
> > Overall, I want to keep my origin rate - weak accept.

---

> > > ### Author Response · Authors · 2022-08-09
> > > **Thanks for the kind suggestions and please consider our more clarifications.**
> > >
> > > We appreciate the careful review of our work and responses. We would still want to clarify some points to address the reviewer's concerns:
> > >
> > > **For question-1:** We realize that there is no conflict between your concern and our observation of reverse distillation: this concern on comparing with architecture-based adaptive distillations is like an apple and orange comparison for "a better teacher":
> > >
> > >
> > > 1. It is true that a good teacher adapts to the student both in terms of model architecture and optimization. We argue that teachers with reverse distillation perform better than original teacher is from the optimization perspective rather than from the model architecture perspective.  These are two different distillation scenarios: the model architecture of the teacher is fixed or optional.  Actually,  teacher's architecture in most offline and online KD is fixed and our observations about reverse distillation are presented in this scenario. While the teacher model/assistant in TAKD and other search-based approaches is optional and adapt to the student model from an architectural perspective. Considering that the two methodologies are different technical routes, the reverse distillation is also orthogonal to TAKD and other search-based approaches. That is to say, adapted teacher assistants in TAKD trained by reverse distillation also could obtain additional distillation gains.
> > >
> > > 2. In theory, we never claims that optimization-based adaptive distillations is better than the architecture-based adaptive distillations. In applications, for the scenario with the pre-trained teacher model given, the teacher-search approach requires additional search and training of a more adapted teacher-assist model, which requires more training pre-training than optimization-based adaptive distillations.
> > >
> > > 3. We thank you very much for this very kind and valuable suggestion. We would like to make efforts to improve future work based on your suggestions.
> > >
> > >
> > > **For question-3:**  The training cost of adapter-based methods is $\times$1.65 than baseline of ResNet-20 (see the Table 1 in the revision).
> > >
> > >
> > > **For question-4:**  These improvements based on the stronger baseline are comparatively sufficient for object detection tasks.  We would do our best to optimize our general SHAKE for particular tasks in future work.
> > >
> > >
> > > **Finally:** Thank you again for the valuable feedback and patience！We would appreciate your reading these more clarifications and considering improving your score ~~~

---

> ### Author Response · Authors · 2022-08-07
> **Responses to the review of Reviewer txDD-Part-II**
>
>
>
> ---
>
> **Q2:** About the teacher-assistant knowledge distillation.
>
> **A2:** In sharp contrast to teacher assistant (named TAKD[28] in our paper), our SHAKE has clear differences in architecture design, optimization manner and better performance regarding accuracy-cost trade-off (see our detailed comparison of in the introduction [Line 82-85], related work [Line 295-301], method [Line 153-168] and experiments in Table 3/Figure 3).
>
> 1. SHAKE identifies the key role of reverse distillation in fixing the teacher-student gap and efficiently introduced it to the offline KDs, to the best of our knowledge, which are not achieved in prior work. This attempt bridges online/offline training fashions and has significant impact on improving the efficiency and wide application of distillation framework.
>
> 2. The proxy teacher mutually optimized with student is our alternative solution to improve efficiency for fine-tuning the original large model. In principle, SHAKE alters the chain of knowledge transfer from pre-trained teacher→student in KD to pre-trained teacher → proxy teacher ↔ student. Teacher assistant [28] only employs middle networks to distill as large teacher → middle teacher → student, which is not optimized for students and requires multi-step training.
>
> 3. On experiments, SHAKE (72.37%) surpasses teacher assistant  (70.78%) with significant  margins for ResNet-18 with single ResNet-34 as teacher on ImageNet (see training curves in Figure 3 [Line 153-168]).
>
>
> ---
>
> **Q3:** About the adapter-based methods.
>
> **A3:** Following the suggestion, we evaluate the adapter-based method as the strategy for efficient fine-tuning of the teacher model. The results indicate that although the adapter-based method achieves a slight improvement compared to fine-tuning the head, it still performs weaker than fine-tuning the full teacher model. Thanks for the suggestion，and we have added this discussion to the Table 1 and  introduction [Line 55-56] in the revision.
>
> =======================================================================
>
> Top-1 accuracy for KD + with reverse distillation under different teacher fine-tuning strategies.  The adapter-based method is performed by adding adaptation layers (Conv-bn-relu) for the output blocks at each stage of the teacher and fine-tuning these adaptation layers.
>
> | teacher         | ResNet-56 | ResNet-110 | VGG-13 | ResNet-56 |
> | --------------- | --------- | ---------- | ------ | --------- |
> | student         | ResNet-20 | ResNet-32  | VGG-8  | VGG-8     |
> | teacher         | 72.34     | 74.31      | 74.64  | 79.34     |
> | student         | 69.06     | 71.14      | 70.36  | 70.36     |
> | KD              | 70.66     | 73.08      | 72.98  | 73.81     |
> | KD+  | 71.66     | 73.89      | 73.51  | 74.41     |
> | KD+(head)       | 71.05     | 73.25      | 73.19  | 74.02     |
> | KD+(adapter)    | 71.34     | 73.55      | 73.32  | 74.19     |

---

> ### Author Response · Authors · 2022-08-09
> **Responses to the review of Reviewer txDD-Part-I**
>
> We sincerely thank the reviewer for the positive comments on our work! Please see our below responses to the concerns one by one. If the reviewer considers our response adequate, we would appreciate it if the reviewer improves the score.
>
> ----
>
> **Q1:** About why an offline teacher works worse.
>
> **A1:** We appreciate the recognition of our interesting observations and this very good question. We make efforts to address your concerns in three fields:
> 1. We understand "the teacher model is optimized for the student model" means that the teacher model is only updated with predictions from the student model. Actually, the offline teacher in KD updates only based on labels and the teacher in DML and SHAKE is optimized with labels and predictions from the student. We evaluate the teacher is optimized with only student's output in the following table. The results show that the teacher models supervised only by students have worse performance than teacher supervised with students and labels. This is intuitively to be understood: on the supervised task, the more parameters in teacher-student network optimized for labels, the more superior accuracy is achieved on the testset. In addition, from the perspective of knowledge transfer, the teacher model only optimized for the student model would lose its learned useful knowledge on the supervised task.
>
> 2. The teachers in almost of all existing distillations are optimized for labels. Based on this default setting, our observations reveal that the optimization for student by reverse distillation plays a key role in reducing the teacher-student gaps. These observations illustrate why the teacher model optimized with reverse distillation work.
>
> 3. The idea "the teacher model is only optimized for the student model" can be implemented by a particular optimization loop. In the following table, we develop a bi-level optimization to implement the teacher's individual optimization for the student model by slicing part of the validation set data from the training data set. In particular, the outer-loop optimization is the distillation training of the student model, and the inner-loop  optimization is the updating of the teacher's weights on the validation set based on the student's output while fixing the student model. This strategy (teacher-> student (bi-level)) also results in competitive performance gains. This conclusion is also in line with our observation because the teacher network is also optimized for reverse distillation.
>
> =======================================================================
>
> Top-1 accuracy for teacher in KD & DML under training strategies for student (ResNet-20: baseline 69.06%).  "->"  means optimization by CE/KL loss
>
> | Method | pre-teacher-training stage | student-training stage2                        | Top-1 |
> | ------ | -------------------------- | ---------------------------------------------- | ----- |
> | KD     | teacher-> labels           |                                                | 70.66 |
> | KD*    | teacher-> labels           | teacher-> labels & student                     | 71.76 |
> | KD*    | teacher-> labels           | teacher-> student                              | 70.88 |
> | KD*    | teacher-> labels           | teacher-> student (bi-level)                   | 71.55 |
> | DML    | N/A                        | teacher-> labels & student                     | 71.52 |
> | DML*   | N/A                        | teacher-> labels                               | 70.55 |
> | DML*   | N/A                        | teacher-> student                              | 70.01 |
> | DML*   | N/A                        | teacher-> labels; teacher-> student (bi-level) | 71.43 |

---

> ### Author Response · Authors · 2022-08-09
> **Thanks for the positive comments and please consider our latest response！**
>
> We sincerely appreciate the reviewer for the thorough and constructive comments. We are really happy that the novelty of the proposed method was recognized by the reviewer. To address the concerns and requests from the reviewer, we are carefully improving the explanations, claims, experiments, and discussions.
>
> Our detailed rebuttal responses have been updated in the last two days with more discussion and experiments in **A1& A4**. We would appreciate your reading these updated responses and considering improving your score!
>
> Thank you again for your time, feedbacks and patience.

---

### Official Review · Reviewer_zGhB · 2022-07-09

**Rating:** 5
**Confidence:** 4
**Soundness:** 3 good
**Presentation:** 3 good
**Contribution:** 3 good

**Summary:**

Offline distillation can employ existing models yet demonstrates inferior performance than online ones. This paper first empirically shows that the essential factor for their performance gap lies in the reversed distillation from student to teacher, rather than the training fashion. Offline distillation can achieve competitive performance gain by fine-tuning pre-trained teacher to adapt student with such reversed distillation. However, this fine-tuning process still costs lots of training budgets. To alleviate this dilemma, this paper proposes SHAKE, a simple yet effective SHAdow KnowlEdge transfer framework to bridge offline and online distillation, which trades the accuracy with efficiency.

**Questions:**

The conclusion from Table 1 is only based on one dataset. But the performance margin is not large. Could this be generalized?

What is the exact meaning of shadow head? More explaination and why this name is used should be given.

**Limitations:**

Statistical tests to show the performance difference of different approaches are needed.

**Strengths And Weaknesses:**

The proposed method allows the proxy teacher to share the student’s backbone but use an individual shadow head to preserve the diversity of logits representations.

SHAKE (share) presents three benefits: (1) more than 3× training acceleration than DML and fine-tuning whole teacher model. (2) No need for architecture selection costs for proxy teacher. (3) Additional accuracy gains than individual proxy teacher because knowledge inherited by proxy teachers also directly improves the representation of students’ backbone in the weight-sharing process.


In principle, SHAKE alters the chain of knowledge transfer from pre-trained teacher→student in
KD to pre-trained teacher→proxy teacherstudent.

I think the idea is interesting though there are some related works.

The conclusion from Table 1 is only based on one dataset. But the performance margin is not large. Could this be generalized?

What is the exact meaning of shadow head? More explaination and why this name is used should be given.

Statistical tests to show the performance difference of different approaches are needed.

---

> ### Author Response · Authors · 2022-08-02
> **Responses to the review of Reviewer zGhB-Part-III**
>
>
> ---
>
> **Q4:** About related work.
>
> **A4:** Our SHAKE studies an important problem in knowledge distillation (the difference between online and offline KD), and we believes it would make a good impact on the KD community. We would like to clarify the difference with other work from the following three points:
> 1. SHAKE identifies the key role of reverse distillation in fixing the teacher-student gap and efficiently introduced it to the offline KDs, to the best of our knowledge, which are not achieved in prior work. This attempt bridges online/offline training fashions and has significant impact on improving the efficiency and wide application of distillation framework.
> 2. We introduce reverse distillation by defining a proxy teacher to achieve a better accuracy-efficiency trade-off.  To easily understand our approach, we present descriptions of the proxy teacher. In sharp contrast to related work [28], the the proxy teacher in SHAKE has clear differences in architecture design, optimization manner and better performance regarding accuracy-cost trade-off (see our detailed comparison of in the introduction [Line 82-85], related work [Line 295-301], method [Line 153-168] and experiments in Table 3/Figure 3).  The proxy teacher mutually optimized with student is our alternative solution to improve efficiency for fine-tuning the original large model. In principle, SHAKE alters the chain of knowledge transfer from pre-trained teacher→student in KD to pre-trained teacher → proxy teacher ↔ student. Teacher assistant [28] only employs middle networks to distill as large teacher → middle teacher → student, which is not optimized for students and requires multi-step training. On experiments, SHAKE (72.37%) surpasses teacher assistant (70.78%) with significant margins for ResNet-18 with single ResNet-34 as teacher on ImageNet (see training curves in Figure 3 [Line 153-168]).
> 3. In addition, we present sufficient analysis and experiments (see Table 1/Figure 5/Figure 7/Figure 8 and Teacher-student similarity analysis in ablation study [Line 268-Line 280]) to illustrate the role of SHAKE with reverse distillation in enhancing teacher-student similarity. while other adaptive distillations reduces the teacher-student gap via teacher-student architectural search or adaptation, the key designs (e.g., reverse distillation and proxy teacher) in SHAKE are completely not achieved in prior adaptive distillation works (see discussion in the related work [Line 295-301]).
>
>
> ---
>
>
> **Finally,** we hope our response could address the concerns, and we thank the reviewer again for the helpful comments. We are glad to discuss further comments and suggestions.

---

> ### Author Response · Authors · 2022-08-08
> **Responses to the review of Reviewer zGhB-Part-II**
>
> ---
>
> **Q3:** About the statistical tests of different approaches.
>
> **A3:** Thanks for the suggestion. Following the suggestion, we present the statistical tests (mean±std) of different approaches of our three repeat experiments in CIFAR-100 as follows (also updated in Table 2 of the revision). These results indicate that SHAKE has stable performance improvements. Considering the model's stable performance of different runs on the large-scale ImageNet datasets, we report the Top-1 accuracy of one experiment following the most knowledge distillation work in the ML conferences.
>
> =======================================================================
>
> Comparison of results with advanced distillation methods under the same training setting of 240 epochs.  R32x4, R8x4, R50, MV2, Sv1 and Sv2 stand for ResNet32x4, ResNet8x4, ResNet50, MobileNetV2, ShuffleNetV1 and ShuffleNetV2. We report Top-1 mean±std accuracies (%) over 3 runs
>
> | Teacher       | WRN-40-2   | WRN-40-2   | R32x4      | VGG13      | R50        | R32x4      | R32x4      |
> | ------------- | ---------- | ---------- | ---------- | ---------- | ---------- | ---------- | ---------- |
> | Student       | WRN-16-2   | WRN-40-1   | R8x4       | VGG8       | MV2        | SV1        | SV2        |
> | Teacher       | 75.61      | 75.61      | 79.42      | 74.64      | 79.34      | 79.42      | 79.42      |
> | Student       | 73.26      | 71.98      | 72.5       | 70.36      | 64.6       | 70.5       | 71.82      |
> | FitNets       | 73.58±0.13 | 72.24±0.14 | 73.50±0.08 | 71.02±0.09 | 63.16±0.28 | 73.59±0.26 | 73.54±0.22 |
> | SP            | 73.83±0.12 | 72.43±0.14 | 72.94±0.16 | 72.68±0.27 | 68.08±0.09 | 73.48±0.07 | 74.56±0.11 |
> | RKD           | 73.35±0.24 | 72.22±0.25 | 71.90±0.18 | 71.48±0.26 | 64.43±0.08 | 72.28±0.09 | 73.21±0.18 |
> | CRD           | 75.48±0.18 | 74.14±0.22 | 75.51±0.17 | 73.94±0.18 | 69.11±0.14 | 75.11±0.13 | 75.65±0.22 |
> | Review        | 76.12±0.12 | 75.09±0.26 | 75.63±0.25 | 74.84±0.15 | 70.37±0.17 | 77.14±0.22 | 77.78±0.18 |
> | CL            | 74.25±0.12 | 72.63±0.14 | 73.10±0.18 | 71.26±0.22 | 65.76±0.22 | 73.62±0.24 | 73.98±0.23 |
> | AFD           | 73.70±0.14 | 72.37±0.15 | 72.98±0.27 | 70.88±0.25 | 64.93±0.21 | 73.68±0.16 | 74.32±0.18 |
> | ONE           | 74.68±0.24 | 73.43±0.26 | 73.51±0.28 | 72.01±0.27 | 66.26±0.22 | 74.35±0.25 | 75.12±0.21 |
> | KD            | 74.92±0.12 | 73.54±0.15 | 73.33±0.18 | 72.98±0.22 | 67.35±0.24 | 74.07±0.26 | 74.45±0.28 |
> | KD+           | 75.58±0.22 | 74.24±0.22 | 74.91±0.22 | 73.65±0.22 | 68.81±0.22 | 75.21±0.12 | 75.95±0.16 |
> | DML           | 75.33±0.18 | 73.98±0.22 | 74.30±0.24 | 73.64±0.15 | 68.52±0.22 | 75.58±0.18 | 76.44±0.16 |
> | DML+          | 74.83±0.24 | 73.26±0.21 | 73.15±0.25 | 72.86±0.26 | 67.27±0.28 | 74.02±0.15 | 74.32±0.16 |
> | SHAKE         | 76.82±0.12 | 75.62±0.14 | 77.95±0.12 | 74.99±0.18 | 70.18±0.16 | 77.46±0.18 | 78.51±0.16 |
> | KD+FitNets    | 75.12±0.25 | 73.86±0.26 | 74.66±0.24 | 73.22±0.22 | 66.81±0.26 | 74.86±0.28 | 75.15±0.26 |
> | KD+CRD        | 75.64±0.24 | 74.38±0.24 | 75.46±0.25 | 74.29±0.28 | 69.54±0.26 | 75.12±0.25 | 76.05±0.29 |
> | SHAKE+FitNets | 76.91±0.12 | 75.73±0.22 | 78.06±0.20 | 75.15±0.18 | 70.23±0.16 | 77.62±0.16 | 78.69±0.14 |
> | SHAKE+CRD     | 77.17±0.12 | 75.89±0.16 | 78.13±0.18 | 75.26±0.12 | 70.42±0.18 | 77.86±0.16 | 78.82±0.18 |
> | KD+Mixup      | 76.58±0.24 | 76.10±0.22 | 77.07±0.26 | 75.58±0.19 | 71.29±0.22 | 78.22±0.28 | 79.14±0.18 |
> | KD+CutMixp    | 76.81±0.24 | 76.45±0.26 | 76.90±0.24 | 75.50±0.25 | 71.10±0.32 | 77.92±0.19 | 79.53±0.20 |
> | SHAKE+Mixup   | 78.20±0.18 | 77.36±0.16 | 79.45±0.15 | 78.32±0.12 | 73.88±0.16 | 79.52±0.18 | 80.86±0.15 |
> | SHAKE+CutMix  | 78.45±0.22 | 77.53±0.23 | 79.59±0.16 | 78.56±0.18 | 74.25±0.16 | 79.98±0.24 | 81.22±0.20 |
> | KD(AVER)      | 75.22±0.24 | 73.92±0.29 | 74.99±0.23 | 74.07±0.28 | 70.21±0.28 | 76.30±0.25 | 75.87±0.28 |
> | KD(AEKD)      | 75.68±0.24 | 74.24±0.36 | 75.15±0.32 | 74.11±0.23 | 70.47±0.24 | 76.34±0.22 | 75.95±0.22 |
> | SHAKE(AVER)   | 77.32±0.15 | 76.22±0.22 | 78.59±0.28 | 75.60±0.13 | 71.9±0.21  | 78.61±0.22 | 78.98±0.29 |
> | SHAKE(AEKD)   | 77.88±0.14 | 76.69±0.18 | 78.90±0.22 | 76.26±0.19 | 72.38±0.25 | 78.94±0.19 | 79.41±0.18 |
>
> ---

---

> ### Author Response · Authors · 2022-08-09
> **Responses to the review of Reviewer zGhB-Part-I**
>
> Thanks for the constructive comments. We are encouraged that the reviewer found our idea intriguing and the evaluation comprehensive. We would like to address the comments and questions below. If the reviewer considers our response adequate, we would appreciate it if the reviewer improves the score.
>
> ---
>
> **Q1:** About the conclusion from Table 1.
>
> **A1:** We appreciate the recognition of our interesting observations and this very good question. We make efforts to address your concerns in two fields:
> 1. This conclusion is generalized on different datasets. Following your suggestion, we evaluate the reverse distillation on multiple datasets as follows. The results demonstrate that KD with reverse distillation outperforms KD with 3.27% ∼ 4.35% margins and reversed distillation obtains 1.65% ∼ 3.42% relative gains for DML than DML without reverse distillation on fine-grained recognition datasets (e.g., CUB200, MIT67,Dogs and Stanford40).
> 2. On the CIFAR-100 and TinyImageNet, the improvement of reverse distillation (0.8% ~1.5% gains for KD & DML and 1.6% ~ 3.2% gains for baseline)  is already obvious for knowledge distillation methods (usually with 0.2% - 1.8% gains than baseline). Note that  KD & DML themselves are very strong distillation methods and outperforms most of the other distillation methods (e.g., Fitnets, AT, NST, CC, RKD, AB, PKT, FT, VID, etc.) reported in CRD [39].
>
> =================================================================================================
>
> Result of ResNet-18 as a student model under distillation training of ResNet-50 as a teacher model. We report top-1 mean±std accuracy (%) over three runs.
>
>  Method                                 | CIFAR-100  | TinyImageNet | CUB200     | MIT67      | Dogs       | Stanford40 |
> | -------------------------------------- | ---------- | ------------ | ---------- | ---------- | ---------- | ---------- |
> | Baseline                               | 73.80±0.60 | 54.60±0.43   | 51.72±1.17 | 55.00±0.97 | 63.38±0.04 | 42.97±0.66 |
> | KD                                     | 77.03±0.25 | 58.36±0.33   | 61.05±1.05 | 57.69±0.58 | 67.50±0.32 | 51.66±1.32 |
> | DML                                    | 77.42±0.29 | 59.54±0.36   | 64.44±0.92 | 59.83±1.63 | 68.51±0.87 | 54.09±0.47 |
> | DML w/o reversed distillation KL(S->T) | 76.88±0.25 | 58.26±0.55   | 57.86±1.12 | 57.35±0.78 | 66.86±0.22 | 51.36±1.12 |
> | KD w reversed distillation KL(S->T)    | 77.91±0.14 | 59.91±0.61   | 65.39±1.03 | 61.74±0.67 | 70.77±0.20 | 56.00±1.19 |
>
> ---
>
> **Q2:** About the exact meaning of shadow head.
>
> **A2:** The head is named shadow head since it imitates the original teacher model just like its shadow. (see our introduction [Line 77-78]).  In addition,  SHAKE arranges individual heads for multiple teacher models as they each have different shadows. The "shadow" means that like the shadow following the person—very closely associated with each other.
>
> ---

---

> ### Author Response · Authors · 2022-08-09
> **Thanks for the positive comments and please consider our latest response.**
>
> We sincerely appreciate the reviewer for the thorough and constructive comments. We are really happy that the novelty of the proposed method was recognized by the reviewer. To address the concerns and requests from the reviewer, we are carefully improving the explanations, claims, experiments, and discussions.
>
> Our detailed rebuttal responses have been updated in the last days with more discussions in A1& A4. We would appreciate your reading these updated responses and considering improving your score!
>
> Thank you again for your time, feedbacks and patience.

---

### Official Review · Reviewer_Xeu9 · 2022-07-09

**Rating:** 5
**Confidence:** 3
**Soundness:** 2 fair
**Presentation:** 3 good
**Contribution:** 2 fair

**Summary:**

The major contribution of this paper is:
1) This paper proposes a new Knowledge Distillation / Transfer scheme called **SHAKE** (**SH**adow **K**nowl**E**dge) to bridge offline and online knowledge distillation / transfer schemes. The proposed method achieves noticeable improvements in image classification (CIFAR-100, ImageNet) tasks.

**Questions:**

Please see Weaknesses section above for a list of questions. Further please consider answering the following questions:

1) Can the authors include Top-5 accuracies for Tables 2, 3?
2) I believe this is not a theoretical paper, not sure why the answer is [Yes] for questions in Checklist (2).


**Limitations:**

Briefly mentioned in Checklist 1(C).

**Strengths And Weaknesses:**

**Strengths:**
1) This paper is written clearly. It is easy to follow.
2) The proposed SHAKE framework obtains good improvements on image classification (CIFAR-100, ImageNet) benchmarks.

**Weaknesses:**
1) My major concern is that this paper lacks analysis. This paper talks about student-aware logits, but no analysis / visualization / quantitative results are provided to add merit to “student-aware” logits claim except the final KD accuracy. Given that this is purely an empirical paper, I’m not convinced regarding the student-aware logits claim. I’m happy to see that the proposed method obtains improvement, but this paper can significantly benefit from good analysis (There are ablation experiments done for CIFAR-100 which I feel is not about student-aware logits). Some useful analysis for “student-aware” logits:
- Given the proposed method shares the backbone and uses different heads for teacher and student, penultimate layer visualization / analysis could be useful (See [1, 2, 3, 4]) to understand the logits learnt by the teacher/ student using SHAKE.

2) Object detection procedure lacks technical details:
- For Object detection, classification is treated as a multi-label classification problem rather than a multi-class classification problem. Is KL divergence used for multi-label distillation for Object detection?
- How did the authors combine ground truth labels for bounding box, objectness score with soft-label predictions from the teacher? I.e.: If the teacher predicts 10% probability for an incorrect class (so no bounding box information in ground-truth), how did the authors perform KD subsequently? Can the authors provide loss function details?


**Minor concerns:**
1) Figure 6 is not an attention map: Given a sample and target class (usually ground truth), it shows pixel-level visual explanations. Showing the probabilities for the correct class across all 5 methods for these 3 samples could be more informative to understand Figure 6 (you can still get pixel-level explanations for wrong predictions, See [5]).

Empirical results of this paper are interesting. But given the lack of analysis, I feel that the weaknesses of this paper outweigh the strengths. I’m happy to change my opinion based on the rebuttal.


References

[1] Müller, Rafael, Simon Kornblith, and Geoffrey E. Hinton. "When does label smoothing help?." Advances in neural information processing systems 32 (2019).

[2] Shen, Z., Liu, Z., Xu, D., Chen, Z., Cheng, K. T., & Savvides, M. (2021). Is label smoothing truly incompatible with knowledge distillation: An empirical study. In ICLR


[3] Lukasik, M., Bhojanapalli, S., Menon, A., & Kumar, S. (2020, November). Does label smoothing mitigate label noise?. In International Conference on Machine Learning (pp. 6448-6458). PMLR.

[4] Chandrasegaran, K., Tran, N. T., Zhao, Y., & Cheung, N. M. (2022). Revisiting Label Smoothing and Knowledge Distillation Compatibility: What was Missing?. In International Conference on Machine Learning

[5] Selvaraju, R. R., Cogswell, M., Das, A., Vedantam, R., Parikh, D., & Batra, D. (2017). Grad-cam: Visual explanations from deep networks via gradient-based localization. In Proceedings of the IEEE international conference on computer vision (pp. 618-626).

---

> ### Author Response · Authors · 2022-08-02
> **Responses to the review of Reviewer Xeu9-Part-III**
>
>
> ---
>
> 2. Our SHAKE focuses on presenting a novel general distillation paradigm, tackling teacher-student gaps instead of detection-oriented distillation loss. Thus, we provide simple extensions for SHAKE to the object detection following the general logits distillation. These extensions can also be improved with detection-oriented knowledge distillation methods (see **A4** for the Reviewer txDD).
>
> 3. The point mentioned is a very good concern: How to match the predictions of classification and localization in object detection. Recent object detection methods [R1, R2, R3] are also focusing on mutual supervision for the classification head and regression head.  Following your suggestion, we evaluate SHAKE with regression distillation guided by classification results of teacher-student via weighting loss. The results SHAKE (logits+) in the following table show that this sample strategy can make some improvement.
>
> =======================================================================
>
> Comparison of results of object detection on MS-COCO.
>
> | Model                          | AP    | APL   | APM   | APS   |
> | ------------------------------ | ----- | ----- | ----- | ----- |
> | Faster-R-CNN-R101 [T]          | 42.04 | 54.6  | 45.55 | 25.22 |
> | Faster-R-CNN-R50 [S]           | 37.93 | 49.1  | 41.14 | 22.44 |
> | KD (logits)                    | 38.35 | 49.48 | 41.8  | 22.73 |
> | Review (feature)               | 40.36 | 52.87 | 43.81 | 23.60  |
> | SHAKE (logits)                 | 38.95 | 50.78 | 42.32 | 22.88 |
>  |SHAKE (logits+)                 | 39.24 | 51.16 | 42.65 | 23.23 |
>
> .
>
> We also have added this discussion to the supplementary material in the revision. Code will be made publicly available.
>
> ---
>
>
> **Q4:** About Top-5 accuracies for Tables 2, 3.
>
>
> **A4:** Most distillation methods always report Top-1 accuracy on CIFAR-100 and ImageNet in the literature, the Top-5 accuracy gain usually has a similar trend with Top-1 accuracy. Following your suggestion, we present the Top-5 accuracy for some of the methods in the ImageNet experiments as follows:
>
> =====================================================================================
>
> | Teacher  | Student   | Acc   | Teacher | Student | KD    | ESKD  | ATKD  | ONE   | DML   | CRD   | SHAKE | SHAKE* |
> | -------- | --------- | ----- | ------- | ------- | ----- | ----- | ----- | ----- | ----- | ----- | ----- | ------ |
> | ResNet34 | ResNet18  | Top-1 | 73.40   | 69.75   | 70.66 | 70.89 | 70.78 | 70.55 | 71.03 | 71.17 | 72.37 | 72.73  |
> | ResNet34 | ResNet18  | Top-5 | 91.42   | 89.07   | 89.88 | 90.06 | 89.99 | 89.59 | 90.28 | 90.32 | 91.25 | 91.86  |
> | Teacher  | Student   | Acc   | Teacher | Student | KD    | AT    | RKD   | OFD   | DML   | CRD   | SHAKE | SHAKE* |
> | ResNet50 | MobileNet | Top-1 | 76.16   | 70.13   | 70.68 | 70.72 | 71.32 | 71.25 | 71.13 | 71.40 | 72.96 | 73.42  |
> | ResNet50 | MobileNet | Top-5 | 92.86   | 89.49   | 90.30 | 90.03 | 90.62 | 90.34 | 90.22 | 90.42 | 91.35 | 91.42  |
>
> ---
>
> **Q5:** About the attention map.
>
>
> **A5:** Thanks for the suggestion. We have already fixed it in the revision (see Line 274-275).
>
> ----
>
> **Q6:** About the answer in Checklist.
>
> **A6:** Thanks for the suggestion. We already changed it to [No] in the revision.
>
> ---
>
> **References:**
>
> [R1]  Zhang H , Fromont E , Lefevre S , et al. Localize to Classify and Classify to Localize: Mutual Guidance in Object Detection, ACCV2020.
>
> [R2] Gao Z ,  Wang L ,  Wu G . Mutual Supervision for Dense Object Detection, ICCV2021.
>
> [R3] Feng C ,  Zhong Y ,  Gao Y , et al. TOOD: Task-aligned One-stage Object Detection, ICCV2021.
>
> ---
> **Finally,** we hope our response could address the concerns, and we thank the reviewer again for the helpful comments. We are happy to discuss further comments and suggestions.

---

> ### Author Response · Authors · 2022-08-02
> **Responses to the review of Reviewer Xeu9-Part-II**
>
>
> ---
>
> **Q3:** About the technical details for object detection.
>
> **A3:** Thanks for your kindness and this thoughtful question. We make efforts to address your concerns in three fields:
> 1. Yes. Conventional detectors (e.g., Faster R-CNN and RetinaNet) independently compute classification loss and regression loss and then optimize their sum. Following these methods and general logits distillations, our SHAKE conducts distillation on the classification and regression output predictions. For classification, we exploit the KL divergence for multi-label distillation. For regression, our SHAKE minimizes the distance of bounding box between teacher-student. The two distillation loss terms are combined with summation, and the details are implemented as follows.
> ```python
> def compute_shake_kd_output_loss(pred, teacher_pred, model, kd_loss_selected="kl", temperature=20, reg_norm=None):
>     t_ft = torch.cuda.FloatTensor if teacher_pred[0].is_cuda else torch.Tensor
>     t_lcls, t_lbox, t_lobj = t_ft([0]), t_ft([0]), t_ft([0])
>     h = model.hyp  # hyperparameters
>     red = 'mean'  # Loss reduction (sum or mean)
>     KDboxLoss = nn.MSELoss(reduction="none")
>     if kd_loss_selected == "l2":
>         KDclsLoss = nn.MSELoss(reduction="none")
>     elif kd_loss_selected == "kl":
>         KDclsLoss = nn.KLDivLoss(reduction="none")
>     else:
>         KDclsLoss = nn.BCEWithLogitsLoss(reduction="none")
>     KDobjLoss = nn.MSELoss(reduction="none")
>     # per output
>     for i, pi in enumerate(pred):  # layer index, layer predictions
>         # t_pi  -->  torch.Size([16, 3, 80, 80, 25])
>         t_pi = teacher_pred[i]
>         # t_obj_scale  --> torch.Size([16, 3, 80, 80])
>         t_obj_scale = t_pi[..., 4].sigmoid()
>         # zero = torch.zeros_like(t_obj_scale)
>         # t_obj_scale = torch.where(t_obj_scale < 0.5, zero, t_obj_scale)
>
>         # BBox
>         b_obj_scale = t_obj_scale.unsqueeze(-1).repeat(1, 1, 1, 1, 4)
>         if not reg_norm:
>             t_lbox += torch.mean(KDboxLoss(pi[..., :4], t_pi[..., :4]) * b_obj_scale)
>         else:
>             wh_norm_scale = reg_norm[i].unsqueeze(0).unsqueeze(-2).unsqueeze(-2)
>             # pxy
>             t_lbox += torch.mean(KDboxLoss(pi[..., :2].sigmoid(), t_pi[..., :2].sigmoid()) * b_obj_scale)
>             # pwh
>             t_lbox += torch.mean(
>                 KDboxLoss(pi[..., 2:4].sigmoid(), t_pi[..., 2:4].sigmoid() * wh_norm_scale) * b_obj_scale)
>
>         # Class
>         if model.nc > 1:  # cls loss (only if multiple classes)
>             c_obj_scale = t_obj_scale.unsqueeze(-1).repeat(1, 1, 1, 1, model.nc)
>             if kd_loss_selected == "kl":
>                 kl_loss = KDclsLoss(F.log_softmax(pi[..., 5:] / temperature, dim=-1),
>                                     F.softmax(t_pi[..., 5:] / temperature, dim=-1)) * (temperature * temperature)
>                 t_lcls += torch.mean(kl_loss * c_obj_scale)
>             else:
>                 t_lcls += torch.mean(KDclsLoss(pi[..., 5:], t_pi[..., 5:]) * c_obj_scale)
>
>         t_lobj += torch.mean(KDobjLoss(pi[..., 4], t_pi[..., 4]) * t_obj_scale)
>     t_lbox *= h['giou'] * h['kd']
>     t_lobj *= h['obj'] * h['kd']
>     t_lcls *= h['cls'] * h['kd']
>     bs = pred[0].shape[0]  # batch size
>     mkdloss = (t_lobj + t_lbox + t_lcls) * bs
>     return mkdloss
>
> def ft(x):
>     return F.normalize(x.pow(2).mean(2).mean(2).view(x.size(0), -1))
>
> def at(x):
>     return F.normalize(x.pow(2).mean(1).view(x.size(0), -1))
>
> def at_loss(x, y):
>     return (at(x) - at(y)).pow(2).mean()
>
> ```
> ---

---

> ### Author Response · Authors · 2022-08-08
> **Responses to the review of Reviewer Xeu9-Part-I**
>
> Thanks for the valuable comments on our work. We have tried our best to address all the concerns in the last few days. We would really appreciate it if the reviewer considers increasing the score. Please see our responses below one by one：
>
> ---
>
> **Q1:** About the analysis of “student-aware” logits.
>
> **A1:** We appreciate the recognition of our interesting observations and this very good question. We make efforts to address your concerns in three fields:
> 1. Following the suggestion, we add correlation visualization of student and teacher logits  (Figure 8 and Line 278-280)  in the revision.  The visualization clearly demonstrates that teacher in SHAKE have a better correlation with student than KD, which provably affirms the student-aware logits of our SHAKE.
>
> 2. We would like to clarify that our teacher-student gap (KL divergence between their outputs logits) actually demonstrates the “student-aware” logits (see Table 1/Figure 5/Figure 8 and Teacher-student similarity analysis in ablation study [Line 268-Line 280]).  For instance, Figure 5 presents KL divergence between their outputs logits of ResNet with different depths as teachers (i.e., ResNet-110, ResNet-56, ResNet-44 and ResNet-32) and the ResNet-20 as student. The results show that “student-aware” logits from SHAKE always give a higher similarity than KD and DML, resulting in superior performance under teacher models of different depths.
>
> 3. Besides informative visualizations and analysis about teacher-student similarities in SHAKE, we also understand why SHAKE works by presenting training curves in Figure 3 (Line 153-168). SHAKE enjoys a more stable optimization behavior compared to other distillation methods.
>
> ---
>
> **Q2:** About the penultimate layer analysis.
>
> **A2:** Following the suggestion, we add penultimate layer visualization (Figure 7 and Line 278-280) in the revision. The visualization illustrates that comparing with KD, applying SHAKE training helps learn more scattered embeddings, which provably affirms the advantage of our SHAKE. Thanks for the suggestion, and we have cited relevant papers [R1, R2, R3, R4] in the revision.
>
> ---

---

> > ### Comment · Reviewer_Xeu9 · 2022-08-08
> > **Reply**
> >
> > Thank you authors for the great effort on the rebuttal. Authors have addressed my concerns to some extent and I have increased my recommendation.

---

> > > ### Author Response · Authors · 2022-08-08
> > > **Thanks for improving the score and recognition of our interesting work!**
> > >
> > > Thanks again for taking the time to review our paper! We are glad that our rebuttal addresses your concerns. All these helpful suggestions will be involved in the revision.

---

### Meta-Review · Area_Chair_J3Kw · 2022-08-24

**Recommendation:** Accept
**Confidence:** Certain

**Metareview:**

This paper addresses an interesting point: the reversed distillation from student to teacher matters the KD performance. Based on this finding, the authors propose SHAKE to make the best of the worlds of offline and online KD. The technical contribution is significant to the KD community and the authors also provide comprehensive SOTA results on various datasets and model architectures. During rebuttal, the authors addressed most of the reviewers' concerns. AC appreciates the discussions and recommends accept.

**Award:**

No

---

### Decision · Program_Chairs · 2022-09-14

Accept